# Patient experiences with liraglutide for obesity and binge eating disorder–A qualitative study

Ingrid Sørdal Følling[1,2]*, Stine Larsen Reigstad[1], Åsne Ask Hyldmo[1,2], Anne-Sofie Helvik[3]

1 Department of Clinical and Molecular Medicine, Faculty of Medicine and Health Sciences, Norwegian University of Science and Technology, Trondheim, Norway, 2 Centre for Obesity Research and Innovation, Clinic of Surgery, St. Olavs Hospital, Trondheim University Hospital, Trondheim, Norway, 3 Department of Public Health and Nursing, Faculty of Medicine and Health Sciences, Norwegian University of Science and Technology, Trondheim, Norway

* ingfoll@gmail.com

## Abstract

### Introduction

Obesity is a growing health concern and a known risk factor for binge eating disorder (BED). BED is characterized by episodes of overeating with a loss of control, often leading to psychological distress. In some cases, medication may be recommended to manage depressive symptoms and support weight loss. Liraglutide, a glucagon-like peptide-1 (GLP-1) analog, works by targeting the brain's reward system to reduce psychological stress and enhance feelings of fullness. However, there is limited research on the use of liraglutide for patients with both obesity and BED. This qualitative study aimed to investigate how patients with these conditions experience treatment with liraglutide.

### Methods

A qualitative design with individual semi-structured, in-depth interviews was employed. Eight informants aged 25–60 years were interviewed, and data were analyzed using systematic text condensation.

### Results

Two main themes, each with three associated subthemes, emerged. The first main theme was: *The role of food on the expression of BED*, with subthemes: *Food as an emotion regulator, Persistent thoughts on eating and dieting*, and *Emotional and situational triggers for binge eating*. The second main theme was *Experiences with liraglutide in managing BED*, with subthemes: *Meeting emotional and physical needs, Reducing thoughts about food*, and *Decreasing triggers for eating*. The experiences

**Data availability statement:** Data cannot be shared publicly due to ethical and legal restrictions related to participant confidentiality. The dataset consists of sensitive, in-depth qualitative interviews with a small number of participants diagnosed with binge eating disorder. Furthermore, participants consented to the use of their data solely within the scope of the present study, as approved by the Regional Committee for Medical and Health Research Ethics (REK) in Central Norway. Therefore, the dataset cannot be made publicly available. Researchers who meet the criteria for access to confidential data may request access to the dataset by contacting the institutional representative: kontakt@ikom.ntnu.no. For questions related to data privacy or ethical oversight, inquiries may be directed to: Thomas Helgesen (Data Protection Officer, NTNU) [thomas.helgesen@ntnu.no].

**Funding:** The author(s) received no specific funding for this work.

**Competing interests:** The authors have declared that no competing interests exist.

with medication (main theme 2) influenced the expression of BED (main theme 1) as informants reported that liraglutide impacted their BED symptoms.

## Discussion

Findings suggest that patients with obesity and BED found liraglutide helpful in addressing emotional and physical needs, enhancing emotional well-being, social interactions, and overall quality of life. Further qualitative research is needed to explore the long-term impact of liraglutide on emotional and behavioral changes in this population.

## Introduction

The prevalence of obesity has increased globally over the past 40 years [1]. Obesity results from an imbalance between energy intake and expenditure [2], shaped by gene-environment interactions [3]. It is associated with multiple comorbidities and may contribute to the development of Binge Eating Disorder (BED) [4].

BED is the most common eating disorder, characterized by episodes of consuming large amounts of food with a sense of loss of control, without compensatory behaviors [4]. Complications associated with BED include depression, insomnia, headaches, menstrual problems, muscle and joint pain, cardiovascular diseases, gallbladder disease, and other digestive issues [5]. The co-occurrence of obesity and BED is associated with reduced quality of life and poorer mental health outcomes [6]. People with obesity and BED often start dieting early [7] and experience frequent weight fluctuations and yo-yo dieting [8]. Dieting is a significant causal factor in pathological binge eating, as it both stimulates binge eating episodes and serves as a reaction to them [9]. Furthermore, negative emotions such as anger, frustration, and tension, as well as psychological distress such as anxiety and depression, and maladaptive emotion regulation can trigger binge eating [10]. A BED episode can occur because of shame over the amount of food consumed, leading to nausea and feelings of depression or guilt afterward [11]. This suggests that people binge eat as a strategy to down-regulate negative emotions, likely due to a lack of more adaptive strategies [12].

Given the complex psychological and physiological mechanisms involved, treatment for BED typically includes psychotherapy, with pharmacological options added when needed [13]. Currently, lisdexamfetamine is the only medication approved for BED [14]. Some antidepressants such as Selective Serotonin Reuptake Inhibitors, have been shown to effectively reduce binge-eating frequency, but have not achieved the desired effect on weight loss [15] and has not been approved for this indication. Topiramate, an antiepileptic medication, [16] has shown promise in severe cases of BED by enhancing the effects of psychotherapy, reducing binge-eating frequency, promoting weight loss, and serving as part of combination therapy for long-term weight management [17]. The Glucagon-like peptid-1 (GLP-1) receptor agonist liraglutide affects hunger, since receptor agonists reduce activation in the brain's

appetite and reward areas, such as the parietal cortex, insula, putamen, and orbitofrontal cortex [18]. Its main effect is that it delays gastric emptying and increases the feeling of satiety [19], and reduces the appeal of highly palatable foods [20]. In patients with obesity, liraglutide has consistently demonstrated significant effects in reducing food intake and body weight [21]. Several studies suggest that liraglutide may reduce binge eating for patients diagnosed with BED [22–25], but most research has examined either obesity or BED in isolation. However, there is limited research exploring patients lived experiences with liraglutide when both obesity and BED are present. This study therefore aimed to explore how patients with comorbid obesity and BED experience the use of liraglutide as part of their treatment.

## Materials and methods

### Design

A qualitative design with individual semi-structured in-depth interviews was used.

### Recruitment and sample

Patients were selected based on the following inclusion criteria:

- ≥ 18 years old

- Diagnosed with obesity and BED (criteria as described in the Diagnostic and Statistical Manual of Mental Disorders (DSM-5).

- Using Liraglutide or previously used it.

No formal exclusion criteria were defined, as participants were recruited from an existing patient group already receiving liraglutide treatment and selected based on inclusion criteria.

From 2017 to 2021, the Obesity Clinic at St. Olavs Hospital referred 72 patients to two different pilot projects aimed at treating BED at psychiatric outpatient clinics in Central Norway. The clinics had conducted treatment for BED through two different psychotherapy treatments: psychoeducation or cognitive-behavioral therapy for eating disorders (CBT-E). Psychoeducation involves educating patients with mental disorders about their illness. This includes the causes, progression, consequences, prognosis, and treatment of the illness [26]. CBT-E is an enhanced version of CBT suitable for all eating disorders. It assumes that preoccupation with weight, body shape, and control over food intake are important maintaining factors in an eating disorder [27].

In the present study, information letters were sent to all these patients on the 30th of January 2023, independent of whether they received CBT-E or Psychoeducation. Those interested in participating contacted the study coordinator via phone or email to schedule interview times and locations according to their preferences. Eight informants between 25 and 60 years old wanted to participate, seven women and one man. The recruitment ended on the 28th of March 2023. Four had previously received CBT-E and the four Psychoeducation. Three informants had previously undergone metabolic bariatric surgery. Four were employed or in education, and four were on disability or work assessment allowance. Regarding marital status, five were cohabiting and three were single. Four informants were using the anti-obesity medication liraglutide at the time of the interview, while four had discontinued its use. Among those who had stopped, usage durations ranged from three months to one year. One discontinued due to side effects, and three stopped due to surgery or pregnancy. Two of these informants planned to restart the medication as soon as possible.

### Data collection

Interviews were conducted at St. Olavs Hospital, Norway in February and March 2023. Recordings were made of all interviews. An interview guide was developed with questions exploring experiences with overeating, including its onset, characteristics of episodes, and perceived causes. The questions further focused on the use of liraglutide, the personal

experiences with the medication, including its effectiveness and any changes in overeating or weight loss. Interview durations ranged from 50 to 92 minutes, with an average of 69 minutes.

## Analysis

All interviews were transcribed verbatim and Malterud's systematic text condensation, a form of thematic analysis, was used to analyze the interview data [28]. The step-by-step analysis was primarily conducted by the second author, with input from the first and third authors during preliminary stages. The first step involved reviewing the transcriptions multiple times to identify initial impressions and themes. Keywords and notes were used to outline commonalities. The second step focused on identifying meaningful units, systematically reviewing each line to link these units to the preliminary themes. In the third step, meaningful units were categorized into codes, which were then grouped into thematic categories. Codes were refined, and similar ones merged for clarity. Finally, the fourth step involved summarizing and developing key themes based on insights from the code groups. To ensure consistency and trustworthiness, the first and third authors were involved in reviewing transcripts and preliminary impressions early in the process. Coding and theme development were discussed iteratively in several rounds among the first, second, and third authors, allowing for refinement and confirmation of interpretations. The last author reviewed the final analysis to ensure coherence across the material. This collaborative approach strengthened the credibility and reliability of the findings.

The interviews were reviewed once more to validate the analysis, with minor adjustments made to ensure clarity and accuracy. The final thematic structure reflects a joint interpretation grounded in the shared analytic process.

## Ethical considerations

The study was approved by the Regional Ethics Committee (REK nr: 561190). Prior to the interviews, consent forms were read through and signed by both parties. Informants were provided with verbal information about the study's purpose, voluntary participation confidentiality and anonymization and their right to withdraw from the project at any time. Data were stored securely on a password-protected server at St. Olavs Hospital in accordance with data protection regulations. Only the research team had access to the data. Consent forms were stored separately from interview material to further protect confidentiality.

## Results

From the informants' stories, two main themes with three associated subthemes eachemerged. Main Theme 2 influenced Main Theme 1, as the informants' experiences with medication impacted the expression of BED (see Fig 1).

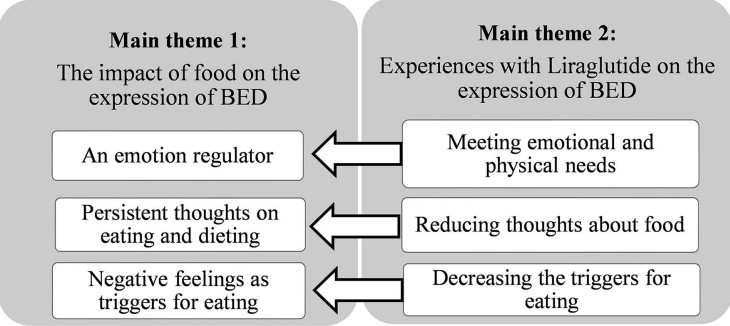

**Fig 1. The main themes and associated subthemes of the results.**

## The role of food on the expression of BED

The informants reported varying experiences with BED at the time of the interview. All participants provided detailed accounts of their condition, with descriptions dating back to when the disorder was at its most severe. They reported consuming high-calorie foods during BED episodes, such as candy, pastries, ice cream, cakes, and chips. Some also mentioned overeating regular meals or sandwiches. BED episodes could occur multiple times a day, over consecutive days, or extend through entire days perceived as continuous episodes.

**Food as an emotion regulator.** The narratives of the informants revealed that they began using food for comfort early in life (see Table 1).

The informants recounted experiences of not having their emotions acknowledged and not learning how to manage or regulate their feelings as children. Some mentioned that anger and irritation were the primary emotions they encountered at home. They also described being deprioritized, left alone, overlooked, and insecure. One informant recalled eating accompanied by vomiting as early as three years old. Informants described how food could calm the body, have a numbing effect, provide comfort, and offer a sense of security or escape from pain and negative thoughts.

The primary emotions they associated with binge eating were anxiety, restlessness, and frustration. Many noted that the effects and emotional dampening from food occurred when their body began to hurt, or they felt physical discomfort or nausea. One stated that binge eating often led to (not self-induced) vomiting, which marked the point when the desired effect was felt.

**Persistent thoughts on eating and dieting.** The informants highlight that food occupies a significant amount of mental space and cognitive capacity. They report extensive planning around grocery shopping and frequently thinking about their next meal. One participant described these thoughts as, *"Food sits in the middle of my forehead."* Additionally, some experience a compulsion to finish all the food on their plate, with one informant describing a sensation of chest pressure if the plate is not emptied. Others recount hiding food and eating in secret.

**Table 1. Informants' quotes on emotional regulation through food.**

| |
|---|
| *"The first memory I have is when I'm about four years old, and there's been drinking at home for several days, and a lot of violence. Then I get a bag of potato chips in my hand. I can remember the physical feeling I get from what I eat then... There's something in my body that calms down (...) It's like that feeling of safety."* |
| *"I felt unconscious. I think that's kind of the point for me. Not having those thoughts about how hopeless I am, how ugly I am, how big I am and all that."* |
| *"Food can also be used to just silence the thoughts, if that makes sense. That you just take it, and it's like... calming down both body and mind with it."* |
| *"It's anxiety, but it's so uncomfortable. And I know that if I eat sugar, it calms down right then."* |
| *"And it's kind of to comfort myself. That I deserve this now because I'm so tired."* |
| *"The only solution I have is to get something that can numb it... yes, call it the pain, or frustration, or maybe the shame I have the most over myself."* |
| *"I get irritated, and then I eat in such large amounts that the pain in my body is what stops me. I eat until it hurts, which brings a sense of calm. When the pain subsides, I start eating again."* |
| *"Feeling highly anxious, like it is a thousand ants inside the body, with overeating is then a relief."* |
| *"I feel unable to stop the episode once it starts. During binge eating, I feel like I am absent or enter a blackout or trance-like state. It is like being unconscious during the act."* |
| *"It's kind of like taking a calming pill, where I kind of feel my body knows that now I'm meeting those needs that I need (...) I think, in a way, it's like my inner child is kind of calmed down a bit then, in a way."* |
| *"But then it was a way to numb the pain and the thoughts. You eat until you can't eat anymore, so you don't have to think."* |
| *"It's especially that feeling of safety then (...) And it's also been a way to regulate all the emotions then, after a while. Because for me, food can calm, but it can also give a boost to get a little energy."* |

Daily cravings for food or sweets can impair concentration and lead to disengagement from conversations. Many informants discuss the struggle of balancing food intake, often accompanied by persistent thoughts like, *"I shouldn't."* Some find it particularly challenging to handle healthy and nutritious foods, describing it as mentally overwhelming. For instance, one participant said, *"If I am going to make oatmeal, I start to gag. I wonder if it's because I associate all healthy food with weight loss."*

Most informants recall periods characterized by restrictive eating and attempts at weight loss. These efforts often involved strict rules, diets, and categorizing foods into "yes" or "no" groups. They describe an intense focus on their bodies and a fixation on using every possible method to lose weight. Their accounts include fasting or consuming minimal food for several days, frequent weighing throughout the day, and skipping social events or birthdays to avoid eating.

One participant noted that stepping away from weight reduction efforts led to less binge eating, fewer intrusive thoughts about food, and a greater sense of control. Conversely, attempting to maintain a caloric deficit or avoid certain foods often resulted in increased binge eating and loss of control. Failing to adhere to a diet or achieve weight-related goals could also intensify negative thoughts, further fueling the urge to binge eat.

**Emotional and situational triggers for binge eating.** Informants report struggles with low self-esteem, feelings of disappointment in themselves, a lack of a sense of achievement, and frequent comparisons with others. Fatigue, depression, family conflicts, a stressful work environment, and interactions with social services are cited as significant triggers for binge eating. Stigma emerges as a particularly powerful trigger, especially comments from others about weight and appearance. Informants' express fears of judgment and describe experiencing negative self-thoughts, such as feeling useless or disgusting. One participant recounted remark like, *"Oh my god, why are you so big? You work out all the time,"* and *"You look pretty now, but if you were a bit thinner, you'd be much prettier."* Another described how riding the bus could evoke feelings of taking up too much space or bumping into others, which could prompt the urge to eat as a coping mechanism. Shame associated with eating in front of others also leads to behaviors like undereating out of fear of judgment. One participant shared, *"When I lived with a partner for a few years, I had to hide food to eat when I was alone. I didn't eat much with others."* Dieting failures frequently triggered feelings of repeated defeat, shame, and uselessness. As one informant explained,

> *"If I somehow missed my diet, ate more than I should have, these thoughts would immediately start: 'Ah, now you're useless again,' right? And when I feel useless, I get the urge to binge eat."*

Informants also discussed negative emotions such as panic, catastrophic future thoughts, fear of losing control over eating, and fear of weight gain. These emotions often led to compensatory behaviors aimed at mitigating the impact of overeating. Such behaviors included exercising excessively, avoiding certain foods, or punishing themselves by not eating. However, these compensatory actions frequently became triggers themselves, perpetuating the cycle of binge eating and maintaining dysfunctional eating patterns.

## Experiences with liraglutide in managing BED

Informants describe varied experiences with liraglutide. Some found it effective initially, though its impact diminished over time, while others reported a sustained positive effect.

**Meeting emotional and physical needs.** Several informants note that the medication helped them achieve quicker satiety, a sense of contentment, calmness, and satisfaction after meals. They share that liraglutide allows their emotional and physical needs to be met without requiring overeating to the point of feeling uncomfortably full, as was typical during binge eating episodes. For instance, one informant remarked that initially, "just one muffin could give the same feeling as overeating."

Some discuss whether it is possible to override liraglutide's effects and continue overeating as before. One informant admitted finding it easy and possible to overeat if desired, while others noted that although their body signals satiety, they

still experience a lingering unease that only dissipates after overeating. Conversely, a few participants shared that the medication made it physically impossible to overeat in the same way as before.

Over time, informants expressed concern that without liraglutide, they might overeat more frequently, especially when daily challenges provoke urges to binge. However, they also highlighted a significant improvement: chest discomfort—a sensation previously associated with overeating—was alleviated even with smaller amounts of food. As one informant summarized their experience:

> "I have overeaten to the point where I've almost had difficulty breathing, and that's not the case at all now. So, in a regular binge eating episode, I'd be stuffed, you know, feeling like I can't eat anymore. Whereas now, it's like I can feel satisfied, even though I'm not stuffed."

**Reducing thoughts about food.**  Many informants report that liraglutide has helped reduce their thoughts about food. They describe a decrease in food preoccupation and cravings, spending less mental energy on meal planning, and experiencing a more relaxed attitude toward food overall. (see quotations in Table 2).

Informants describe developing a sense of indifference toward food, noting that its function has shifted compared to before. This change has made it easier for them to maintain routines and consume healthier meals. Some highlighted a noticeable difference when pausing liraglutide, with previous food preoccupation and mental agitation returning during these breaks.

**Decreasing triggers for eating.**  Many informants view liraglutide as a "helping hand" in managing triggers for binge eating. It has enabled them to process difficult thoughts and emotions without turning to food. With fewer BED episodes, they report reduced feelings of guilt, failure, remorse, and shame. Negative emotions and fears, such as losing control over eating or gaining weight, have become less prominent.

**Table 2.  Quotes about informants' thoughts about food.**

| |
|---|
| "I used a lot of mental effort and capacity to do the right things after I had recovered. When I was ill, I spent a great deal of mental energy planning and thinking about 'what not to eat,' 'now I will binge eat,' or 'damn, I binged last week.' But liraglutide removed the thoughts about food, so suddenly I had a lot of space in my head. It eliminated the cravings for things (...) I didn't have to think so much about it." |
| "Food suddenly became a form of fuel. It didn't become a form of emotional control." |
| "But when I stopped using liraglutide, I felt a strong craving again. A hunger and a focus on food returned. It seems like liraglutide gives me a kind of peace, a sort of food peace (...) It provides a bodily calmness in a way." |
| "The first thing I noticed was a sense of peace around food, similar to how I felt right after surgery—there was no longer an obsession with it. I was so relieved, almost thinking, 'Oh, I have to remember to eat a few meals.' And I was able to eat healthy meals consistently. |
| "I almost think you can compare it to someone with ADHD when they get ADHD medication, it calms down. I don't think about food so much." |
| "I felt that I could distinguish between hunger and cravings. Food moved from the forefront of my mind to the back, so it wasn't the first thing I thought about. And I managed to avoid thinking about three meals ahead. I just thought 'okay, this is what we'll have for dinner, and that's fine' (...) it was relaxing, that's almost the best way to describe it." |
| "In the beginning, it was the complete opposite, I didn't really need to think. I was sort of forced to have control because I got full so quickly, it wasn't a problem. The medication sort of did that for me, took control for me. After a couple of months, I was back in control in a way, and I had to decide for myself how much to eat because I didn't get the same feeling of being completely full." |
| "It feels like a kind of break from negative thoughts. And I avoid much of the guilt associated with feeling like I'm eating too much." |

They also note a reduced need for restrictive eating behaviors, as decreased hunger and increased satiety have lowered overall food intake. One informant shared how the medication helped counter compensatory thoughts related to binge eating and weight gain, allowing them to sustain healthier habits:

*"It also helped with the binge eating disorder, I would say, because the desire to binge eat and punish yourself when you've gained 10 kg is very strong. The urge to start starving or go on a diet was enormous in my case. But I couldn't do that, because then I wouldn't be healthy anymore. So having liraglutide as a sort of helping hand was invaluable, I would say. It allowed me to maintain the routines I had established."*

## Discussion

The study identified two key areas: the role of food in BED and participants' experiences with liraglutide. Food was found to be central to emotional regulation, with emotional and situational triggers for binge eating and persistent thoughts about food and dieting dominating participants' mental space. Conversely, liraglutide helped reduce emotional eating, diminished triggers for overeating, and alleviated obsessive thoughts about food, offering both emotional and physical relief. These findings underscore the emotional drivers behind BED and highlight the potential benefits of liraglutide in addressing these challenges.

### Emotions and emotional regulation in BED

Informants reported turning to food for comfort from an early age, using it to escape negative thoughts and feel secure. They noted that negative emotions often triggered episodes of overeating. This aligns with research indicating that stress and emotional turmoil during childhood can lead individuals to adopt eating as a coping mechanism [29]. While sadness is often linked to BED [12], other emotions like anger, frustration, depression, anxiety, and tension can also act as triggers [8], Similar to the experiences described by our informants, individuals with BED report heightened emotional distress and greater difficulty managing negative emotions, which often lead to episodes of overeating [30]. A meta-analysis found that 69–100% of participants retrospectively reported negative emotions as triggers for their binge eating episodes [31]. Binge eating may initially reduce negative emotions, making it a maladaptive coping strategy that sustains the behavior [10]. Indeed, negative emotions tend to be higher on binge eating days [32] and emotional dysregulation is a key factor in the onset and maintenance of BED [10]. Informants in our study, consistent with findings in the literature, reported that the core issue lies not merely in experiencing negative emotions but in the difficulty of regulating them effectively [12]. Our informants experienced temporary relief from negative emotions during binge eating, but the following shame created a cycle of overeating. Consistent with our findings, research shows that many with BED experience emotional detachment during binge episodes, followed by guilt, shame, and concerns about weight gain [33]. Some report a reduction in negative emotions post-binge eating [34]. Thus, shame often perpetuates the cycle of binge eating [35], quite much the same as our informants describe. This paradox reflects how binge eating, while initially soothing, worsens emotional wellbeing. Both our findings and existing literature suggest that difficulties with emotional regulation, rather than the mere presence of negative emotions, are central to the persistence of BED.

### Liraglutides impact on emotional and physical needs

In our findings, the use of Liraglutide contributed to fewer thoughts about food and a quicker sense of satisfaction, calmness, and contentment after meals. Additionally, emotional and physical needs were met more promptly with liraglutide. This aligns with findings from another study, which found that liraglutide led to greater improvements in hunger, satiety, and reduced preoccupation with food [36]. Research on GLP-1 analogs in patients with BED also supports these findings [22–24]. One study has reported that patients treated with liraglutide had significantly lower scores on the Binge Eating

Scale, with 81% of them transitioning from the "binge eating" category to "non-binge eating" [23]. This improvement is consistent with our informants' experiences, who reported that liraglutide helped curb their binge eating tendencies by reducing both the emotional and physical drive to overeat. Another study found a reduction in binge eating episodes with liraglutide, although it did not show a significant difference compared to the placebo group [22]. This highlights the complexity of treating BED with GLP-1 analogs, as individual responses may vary. However, our findings support the notion that liraglutide helps moderate binge eating episodes by addressing the emotional regulation challenges commonly associated with BED.

A review exploring the effects of GLP-1 analogs on stress-related eating found that long-term use could potentially improve mood, though the exact mechanisms remain unclear due to potential confounding factors like weight loss and improved blood sugar control [37]. The same factors may relate to our informants' reports of increased emotional stability while using liraglutide, as the drug seemed to meet both their physical and emotional needs more effectively. Furthermore, a randomized controlled trial has found that GLP-1 receptor agonists reduce activation in brain regions associated with appetite and reward in response to highly palatable foods [18]. This aligns with our informants' experiences, as they reported fewer intense cravings. Taken together, the participants' accounts suggests that the idea that liraglutide may not only help regulate food intake but also address the emotional triggers of binge eating, highlighting its potential as a complementary treatment approach for BED [38]. While these findings are promising, they are based on a small, qualitative sample and should be interpreted as preliminary. In line with our results and previous literature, participants described a reduction in food-related thoughts and cravings, which they associated with reduced emotional distress and improved ability to concentrate. However, any broader conclusions about emotional regulation or quality of life must be drawn cautiously, and further research is needed to substantiate these impressions in more diverse populations. Additionally, as with any medication, side effects such as nausea, constipation, fatigue, and gastrointestinal discomfort may interfere with daily life. These side effects are well-documented in clinical trials of liraglutide and other GLP-1 receptor agonists [39]. This underscores the importance of weighing the benefits against potential drawbacks when considering Liraglutide for patients with BED and obesity and highlights the need for close follow-up and individualized support throughout treatment.

## Strengths and limitations

A strength of the study is its focus on depth over breadth, which allowed for rich, detailed insights into the experiences of individuals with obesity and BED using liraglutide. While the smaller sample size limited the diversity of perspectives, the study contributes to understanding the nuanced emotional and behavioral changes of the informants. However, the inclusion of predominantly positive experiences with liraglutide may suggest a selection bias, as individuals experiencing favorable outcomes may have been more motivated to participate. This limits the generalizability of the findings to those who may not have responded as well to the treatment.

Additionally, since our interviews were conducted after participants had already started using liraglutide, the lack of pre-treatment data makes it difficult to fully capture the changes in the antecedents and consequences of binge eating episodes. Including pre- and post-treatment interviews or measures would provide a clearer understanding of how liraglutide impacted participants' relationship with food over time.

Another important consideration is the CBT-E or psychoeducation treatments that informants had received. These treatments may have enhanced the overall effect of liraglutide by helping participants develop adaptive coping strategies to regulate emotions and reduce binge eating triggers. Combining pharmacological treatment with psychotherapeutic approaches could lead to more sustainable, long-term improvements in emotional well-being and quality of life by addressing both the physiological and psychological aspects of BED. Future studies could explore how the combination of pharmacological treatment and psychotherapeutic interventions works, to better understand the impact of integrative approaches on patient outcomes.

                                                                 

## Conclusion

Our findings suggest that liraglutide helps patients with obesity and BED by addressing both emotional and physical needs, reducing triggers, and decreasing food-related thoughts, which led to a decline in BED symptoms, aligning with previous research on GLP-1 analogs. Informants also reported improvements in emotional well-being, social interactions, and quality of life. However, due to the exploratory design and small sample size, these results are preliminary and not generalizable.

This study highlights that liraglutide, originally developed for diabetes and weight loss, may also support psychological and behavioral improvements in BED, challenging the notion that pharmacological treatments target only physiological symptoms. These insights point to the potential of integrating GLP-1 receptor agonists into holistic, multidisciplinary care for BED. The findings also have implications for clinical practice and health policy by emphasizing equitable access to medications that benefit both mental health and quality of life. Future research should focus on long-term outcomes and the integration of such treatments into person-centered care. Overall, our exploratory results provide early evidence that liraglutide may impact both physical and emotional aspects of BED, warranting further large-scale studies to inform clinical guidelines.

## Acknowledgments

We would like to acknowledge the patients in this study who shared their stories. Thanks to the Obesity Clinic at St. Olavs that helped in the recruitment of patients to participate.

## Author contributions

**Conceptualization:** Ingrid Sørdal Følling, Stine Larsen Reigstad.

**Data curation:** Stine Larsen Reigstad.

**Formal analysis:** Ingrid Sørdal Følling, Stine Larsen Reigstad, Åsne Ask Hyldmo.

**Investigation:** Ingrid Sørdal Følling.

**Methodology:** Ingrid Sørdal Følling.

**Project administration:** Ingrid Sørdal Følling.

**Supervision:** Ingrid Sørdal Følling, Åsne Ask Hyldmo.

**Validation:** Ingrid Sørdal Følling.

**Writing – original draft:** Stine Larsen Reigstad.

**Writing – review & editing:** Ingrid Sørdal Følling, Åsne Ask Hyldmo, Anne-Sofie Helvik.

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
