## [Decision Letter · Decision Letter 0]

12 May 2025

PONE-D-24-56405Patient experiences with Liraglutide for obesity and binge eating disorder- a qualitative studyPLOS ONE

Dear Dr. Følling,

Thank you for submitting your manuscript to PLOS ONE. After careful consideration, we feel that it has merit but does not fully meet PLOS ONE’s publication criteria as it currently stands. Therefore, we invite you to submit a revised version of the manuscript that addresses the points raised during the review process.

 Please submit your revised manuscript by Jun 25 2025 11:59PM. If you will need more time than this to complete your revisions, please reply to this message or contact the journal office at plosone@plos.org . Please include the following items when submitting your revised manuscript:

We look forward to receiving your revised manuscript.

Kind regards,

Saeed Ahmed, MD, FAPA, FASAM

Academic Editor

PLOS ONE

Journal Requirements:

Additional Editor Comments :

Re: Manuscript ID PONE-D-24-56405

Title: "Patient experiences with Liraglutide for obesity and binge eating disorder – a qualitative study"

Dear Authors,

Thank you for submitting your manuscript to PLOS ONE. We appreciate the opportunity to review your work on this important and timely topic.

Please find below the full reviewer comments. We ask that you carefully address all comments and suggestions as part of your revision. In your resubmission, please include the following:

A revised manuscript with tracked changes or highlights indicating the revisions made.

A clean version of the revised manuscript.

A detailed, point-by-point response letter addressing each reviewer comment individually. Please describe the changes made in response to each point, or provide a rationale if no changes were made.

Your thoughtful and thorough responses will assist in the continued evaluation of your manuscript. We look forward to receiving your revised submission.

Reviewer 1 Comments:

Thank you for the opportunity to review this manuscript. Below are my comments and suggestions for strengthening the manuscript further.

Line 116 – Which antidepressant is FDA approved for BED. Also, medicine like Lisdexamfetamine which is also FDA approved for BED. Include both the medications so that readers may know list of medications that has been approved for BED treatment and may spark interest for future novel therapeutic options.

Line 119 – Topiramate is not FDA approved for BED but can be used as an open label. Nevertheless, it is approved for chronic weight management.

Line 131 – Authors state “limited research exists” on Liraglutide but have not made efforts to identify where gap exists and what current literature states about its use in BED. This will help set up the tone for the entire paper.

Line 141 – Also include exclusion criteria.

Line 209 – Clearly state how data was handled and where data were stored to ensure confidentiality.

Line 217 – How did you analyze your qualitative data—was it thematic, content, or narrative analysis? Explain.

Line 239 – If possible, present thematic analysis findings visually; use diagrams, such as thematic maps, flow charts, etc.

Line 436 – Not all is good with Liraglutide as with any other drugs. Explain limitations and side-effects associated with its use.

Line 509 – Expand conclusion by adding the following points: Highlight how the study adds new insights, challenges assumptions, or fills a gap in research. How can the results inform policy, practice, or interventions? What broader societal insights can be drawn?

Reviewer 2

This article offers a comprehensive overview of the relationship between obesity and Binge Eating Disorder (BED), including their psychological and physiological factors, as well as potential treatment options such as psychotherapy, pharmacotherapy, and the use of Liraglutide. It effectively emphasizes how food acts as an emotional regulator for individuals with BED and how Liraglutide may help alleviate emotional eating and food-related concerns. The discussion on the interaction between pharmacological treatment and psychotherapy is particularly insightful.

However, the study has several limitations: it has a small sample size, lacks generalizability, and does not effectively minimize confounding factors. Despite these issues, the study provides important preliminary evidence regarding the role of Liraglutide in addressing both the emotional and physical aspects of BED. Future research with larger sample sizes, pre- and post-treatment assessments, and long-term follow-up is essential to confirm these findings and explore the full potential of GLP-1 analogs in the treatment of BED.

Reviewer 3 Comments:

The abstract gives a clear and summary of the study, covering the background, methods, results, and key findings. It explains that the study looks at how people with obesity and BED (binge eating disorder) experience using Liraglutide. However, it doesn’t mention the number of participants (8), and it only says the study is qualitative halfway through the abstract. This could confuse readers who might think it’s a clinical trial. It would be better to state early on that it’s a qualitative study using thematic analysis and include the sample size upfront.

The introduction gives useful background information and explains how BED and obesity are related. It also explains why Liraglutide might help. However, the section is a bit too long and repeats similar ideas about emotional eating, dieting, and weight concerns. Also, the main research aim is only mentioned at the end. The introduction would be clearer if the authors explained the study's goal earlier and removed repeated points to make it easier to read.

This section clearly describes how the study was done, including how participants were chosen, what kind of interviews were used, and how the data were analyzed. It also covers the ethical steps taken. However, it doesn’t say whether different researchers checked each other’s coding or how they made sure the themes were reliable. It would be helpful to include information on how they checked for consistency—like if more than one researcher coded the data or if they discussed themes together to make the findings more trustworthy.

The results section is well-supported by rich participant quotes that clearly illustrate each theme. The inclusion of the figure showing the relationship between the two main themes “The role of food on the expression of BED” and “Experiences with Liraglutide in managing BED” adds strong visual clarity. It helps readers understand how Liraglutide affects emotional regulation, food-related thoughts, and binge-eating triggers. This figure strengthens the structure of the results and provides a helpful overview of the study's findings.

The discussion does a good job linking the study findings to past research and explaining how Liraglutide might help with both emotional and physical issues related to BED. But some of the claims, like improving overall quality of life, may be too strong given that the study only included 8 people and used a qualitative design. It would be better to clearly say that these findings are early and can’t be applied to everyone. Highlighting that this is an exploratory study would make the conclusions more balanced.

Reviewers' comments:

Reviewer's Responses to Questions

**Comments to the Author**

1. Is the manuscript technically sound, and do the data support the conclusions?

Reviewer #1: Partly

Reviewer #2: Yes

Reviewer #3: Yes

2. Has the statistical analysis been performed appropriately and rigorously? 

Reviewer #1: I Don't Know

Reviewer #2: No

Reviewer #3: N/A

3. Have the authors made all data underlying the findings in their manuscript fully available?

Reviewer #1: Yes

Reviewer #2: Yes

Reviewer #3: Yes

4. Is the manuscript presented in an intelligible fashion and written in standard English?

Reviewer #1: Yes

Reviewer #2: Yes

Reviewer #3: Yes

5. Review Comments to the Author

Reviewer #1: Thank you for the opportunity to review this manuscript. Below are my comments and suggestions for strengthening the manuscript further.

- Line 116 - which antidepressant is FDA approved for BED. Also, medicine like Lisdexamfetamine which is also FDA approved for BED. Include both the medications so that readers may know list of medications that has been approved for BED treatment and may spark interest for future novel therapeutic options.

- Line 119 - Topiramate is not FDA approved for BED but can be used as an open label. Nevertheless, it is approved for chronic weight management.

- Line 131 - Authors states “limited research exists” on Liraglutide but have not made efforts to identify where gap exists and what current literature states about its use in BED. This is help setup the tone for the entire paper.

- Line 141 - Also include exclusion criteria.

- Line 209 - Clearly state how data was handled and where data were stored to ensure confidentiality.

- Line 217 - How did you analyzed your qualitative data, was it thematic, content, or narrative analysis. Explain.

- Line 239 - If possible present thematic analysis finding visually, use diagrams, such as thematic maps, flow charts etc..

- Line 436 - Not all is good with Liraglutide as with any other drugs. Explain limitations and side-effects associated with its use.

- Line 509 - Expand conclusion by adding following points. Highlight how the study adds new insights, challenges assumptions, or fills a gap in research. How can the results inform policy, practice, or interventions. What broader societal insights can be drawn.

Reviewer #2: This article offers a comprehensive overview of the relationship between obesity and Binge Eating Disorder (BED), including their psychological and physiological factors, as well as potential treatment options such as psychotherapy, pharmacotherapy, and the use of Liraglutide. It effectively emphasizes how food acts as an emotional regulator for individuals with BED and how Liraglutide may help alleviate emotional eating and food-related concerns. The discussion on the interaction between pharmacological treatment and psychotherapy is particularly insightful.

However, the study has several limitations: it has a small sample size, lacks generalizability, and does not effectively minimize confounding factors. Despite these issues, the study provides important preliminary evidence regarding the role of Liraglutide in addressing both the emotional and physical aspects of BED. Future research with larger sample sizes, pre- and post-treatment assessments, and long-term follow-up is essential to confirm these findings and explore the full potential of GLP-1 analogs in the treatment of BED.

Reviewer #3: 1. The abstract gives a clear and summary of the study, covering the background, methods, results, and key findings. It explains that the study looks at how people with obesity and BED (binge eating disorder) experience using Liraglutide. However, it doesn’t mention the number of participants (8), and it only says the study is qualitative halfway through the abstract. This could confuse readers who might think it’s a clinical trial. It would be better to state early on that it’s a qualitative study using thematic analysis and include the sample size upfront.

2. The introduction gives useful background information and explains how BED and obesity are related. It also explains why Liraglutide might help. However, the section is a bit too long and repeats similar ideas about emotional eating, dieting, and weight concerns. Also, the main research aim is only mentioned at the end. The introduction would be clearer if the authors explained the study's goal earlier and removed repeated points to make it easier to read.

3. This section clearly describes how the study was done, including how participants were chosen, what kind of interviews were used, and how the data were analyzed. It also covers the ethical steps taken. However, it doesn’t say whether different researchers checked each other’s coding or how they made sure the themes were reliable. It would be helpful to include information on how they checked for consistency like if more than one researcher coded the data or if they discussed themes together to make the findings more trustworthy.

4. The results section is well-supported by rich participant quotes that clearly illustrate each theme. The inclusion of the figure showing the relationship between the two main themes “The role of food on the expression of BED” and “Experiences with Liraglutide in managing BED” adds strong visual clarity. It helps readers understand how Liraglutide affects emotional regulation, food-related thoughts, and binge-eating triggers. This figure strengthens the structure of the results and provides a helpful overview of the study's findings.

5. The discussion does a good job linking the study findings to past research and explaining how Liraglutide might help with both emotional and physical issues related to BED. But some of the claims, like improving overall quality of life, may be too strong given that the study only included 8 people and used a qualitative design. It would be better to clearly say that these findings are early and can’t be applied to everyone. Highlighting that this is an exploratory study would make the conclusions more balanced.

6. PLOS authors have the option to publish the peer review history of their article (what does this mean? ). If published, this will include your full peer review and any attached files.

**Do you want your identity to be public for this peer review?** For information about this choice, including consent withdrawal, please see our Privacy Policy .

Reviewer #1: **Yes: ** Mohsin Raza

Reviewer #2: No

Reviewer #3: No

---

## [Author Response · Author response to Decision Letter 1]

30 Jun 2025

Response regarding additional requirements

1. We have made the necessary formatting changes to ensure compliance with PLOS ONE’s style requirements, including file naming and manuscript structure, as outlined in the provided templates.

2. Restrictions on Data Availability.

Thank you for emphasizing the importance of data transparency. We fully understand and appreciate PLOS ONE’s data sharing policies. In preparation for this resubmission, and in consideration of PLOS ONE’s data policy, we consulted the Regional Committee for Medical and Health Research Ethics (REK) in Central Norway regarding the possibility of public data sharing. We were concerned about sharing the anonymized data publicly due to the small number of participants which could risk indirect identification. REK expressed reservations due to the sensitive nature of the material with in-depth personal narratives from a small group of participants (n=8). Because of the richness of the content and the limited number of participants, it is not possible to fully guarantee anonymity, even with de-identification procedures. REK also informed that any use of the data beyond what participants originally consented to, including public sharing, would require renewed ethical review and approval. It is outlined in the participants’ consent form that they agreed to the use of their data solely within the scope of this study. Considering these considerations, and in accordance with both ethical and legal obligations, we are unfortunately unable to make the full dataset publicly available. However, we remain committed to transparency and will gladly facilitate access under appropriate conditions.

Additionally, the scientific benefit of making the raw data publicly available is limited, as the dataset consists of verbatim transcriptions of interviews conducted in Norwegian, totaling 118 pages (approximately 580 minutes of audio recordings). Furthermore, the use of Norwegian significantly limits the practical reuse of the data, particularly for international researchers.

However, researchers who are interested in accessing the data may contact the institutional representative to initiate a formal request, subject to approval by REK and the Norwegian Data Accessibility Committee (SIKT). Institutional contact for data access inquiries: Magnus Steigedal, NTNU (Research Responsible) Email: magnus.steigedal@ntnu.no. For questions related to data privacy or ethics:

Thomas Helgesen, NTNU Data Protection Officer. Email: thomas.helgesen@ntnu.no

As requested, we have included a Data Availability Statement in the revised manuscript after the Conclusion, and we have also uploaded a separate document that outlines the restrictions in more detail. Please let us know if you require any further clarifications.

b) We hope for the understanding that the dataset is not uploaded for the reasons above.

2. We have thoroughly reviewed our reference list to ensure that all citations are complete, correct, and current improved formatting to comply with journal guidelines.

A point-by-point response to each reviewer

Reviewer 1

1. Comment: Line 116 – Which antidepressant is FDA approved for BED. Also, medicine like Lisdexamfetamine which is also FDA approved for BED. Include both the medications so that readers may know list of medications that has been approved for BED treatment and may spark interest for future novel therapeutic options.

• Response: We are thankful for this valuable comment. Lisdexamfetamine dimesylate is the only medication currently FDA-approved specifically for BED treatment. We have revised the manuscript according to this and mentioned that while some antidepressants, such as selective serotonin reuptake inhibitors (SSRIs), have shown benefit in clinical trials for BED, no antidepressant has been formally approved by the FDA for this indication. We agree that highlighting these distinctions may help inform readers.

2. Comment: Line 119 – Topiramate is not FDA approved for BED but can be used as an open label. Nevertheless, it is approved for chronic weight management.

• Response: Thank you for this important clarification. We have revised this in the manuscript, noting that Topiramate is serving as part of combination therapy for chronic weight management

3. Comment: Line 131 – Authors state “limited research exists” on Liraglutide but have not made efforts to identify where gap exists and what current literature states about its use in BED. This will help set up the tone for the entire paper.

• Response: We appreciate your helpful comment. We agree that it is important to articulate the gap in the literature regarding Liraglutide use in patients with both obesity and BED. We have revised the text to specify the current state of research. This clarification helps to better justify the need for our study and sets a stronger foundation for the rest of the paper.

4. Comment: Line 141 – Also include exclusion criteria.

• Response: In this study, participants were selected from an existing treatment population based on predefined inclusion criteria. As such, there were no formal exclusion criteria beyond not meeting these inclusion requirements. We have clarified this in the manuscript to ensure transparency.

5. Comment: Line 209 – Clearly state how data was handled and where data were stored to ensure confidentiality.

• Response: Thank you for this important comment. We have revised the ethics section to include a clear description of how the data was handled and stored to ensure participant confidentiality and compliance with data protection regulations.

6. Comment: Line 217 – How did you analyze your qualitative data—was it thematic, content, or narrative analysis? Explain.

• Response: We are grateful for this observation. We appreciate the opportunity to clarify our methodological approach. As described in the Analysis part in Methods section we used Malterud’s Systematic Text Condensation (STC) to analyze our data, which is a pragmatic and structured method for thematic cross-case analysis of qualitative data. We have now explicitly stated that this approach is a form of thematic analysis to ensure clarity for readers.

7. Comment: Line 239 – If possible, present thematic analysis findings visually; use diagrams, such as thematic maps, flow charts, etc.

• Response: Thank you for this suggestion. We agree that visual representations can enhance the clarity and accessibility of qualitative findings. In this study, we chose to present the thematic findings through a quote table (Table 1), which we found particularly effective in illustrating the depth and emotional nuances of the participants’ experiences. These rich, first-person narratives offer insight into how food was used to regulate emotions and form the core of the identified theme. We considered the use of diagrams (e.g., thematic maps), but ultimately found that the direct quotes more authentically conveyed the complexity of the emotional regulation theme than abstract visual representations.

8. Comment: Line 436 – Not all is good with Liraglutide as with any other drugs. Explain limitations and side-effects associated with its use.

• Response: Thank you for this important comment. We agree that it is essential to present a balanced view of liraglutide use. We have revised the manuscript to include a discussion of known limitations and side effects associated with liraglutide, as reported in both the literature and our study participants’ experiences. This addition provides a more nuanced understanding of the medication’s role in treating patients with obesity and BED.

• Comment: Line 509 – Expand conclusion by adding the following points: Highlight how the study adds new insights, challenges assumptions, or fills a gap in research. How can the results inform policy, practice, or interventions? What broader societal insights can be drawn?

• Response: We are grateful for this helpful suggestion. We have expanded the conclusion to highlight how the study contributes new insights into the use of Liraglutide in patients with both obesity and BED. The revised text emphasizes how our findings challenge assumptions that pharmacological treatment only targets physiological symptoms and instead suggest broader emotional and behavioral effects. We have also discussed the implications for clinical practice, policy, and future research, as well as potential societal relevance.

Reviewer 2

1. Comment: This article offers a comprehensive overview of the relationship between obesity and Binge Eating Disorder (BED), including their psychological and physiological factors, as well as potential treatment options such as psychotherapy, pharmacotherapy, and the use of Liraglutide. It effectively emphasizes how food acts as an emotional regulator for individuals with BED and how Liraglutide may help alleviate emotional eating and food-related concerns. The discussion on the interaction between pharmacological treatment and psychotherapy is particularly insightful. However, the study has several limitations: it has a small sample size, lacks generalizability, and does not effectively minimize confounding factors. Despite these issues, the study provides important preliminary evidence regarding the role of Liraglutide in addressing both the emotional and physical aspects of BED. Future research with larger sample sizes, pre- and post-treatment assessments, and long-term follow-up is essential to confirm these findings and explore the full potential of GLP-1 analogs in the treatment of BED.

• Response: We thank the reviewer for highlighting both the strengths and limitations of our work. We recognize the limitations inherent in a small qualitative sample; however, we believe our work provides a crucial first look at how Liraglutide intersect in the lived experience of BED. We have revised the manuscript to make these points clearer and to better frame our study as an exploratory contribution to an emerging field. We agree that our sample is small and that—by design—this qualitative study does not aim for broad generalizability or statistical control of confounders. Instead, our goal was to explore in depth how individuals with BED experience emotional eating and use of Liraglutide. We offer the following points in response:

-Purpose of a qualitative approach:

Our study was intended as an exploratory, hypothesis‐generating investigation rather than a hypothesis‐testing one. By interviewing a small number of participants, we were able to gather rich, nuanced descriptions of how Liraglutide affects emotional regulation around food, body image, and eating behaviors in the everyday lives of people with BED. These detailed first‐person accounts reveal aspects of patient experience that cannot be captured by larger quantitative studies alone. In a field where medication use for BED is relatively new, such in‐depth insights are valuable for identifying themes, language, and patient concerns that can inform the design of future trials.

-Novel contribution to an underexplored area:

Although Liraglutide and other GLP‐1 analogs are increasingly prescribed for weight management, there is very little qualitative research on how these medications interact with patterns of emotional eating in BED. Our interviews uncovered emergent themes, such as changes in relationship to food and anxiety around binges, and the perceived synergy between medication and psychotherapeutic coping strategies—that have not been documented elsewhere. By making these themes explicit, we contribute preliminary evidence about mechanisms of action (both emotional and physical) that can guide the hypotheses and questionnaires of larger, more controlled studies.

-Confounding factors and transferability:

We acknowledge that confounding variables (e.g., variations in psychotherapy style, individual differences in psychiatric comorbidity, prior medication history) are not “controlled” in our design. However, it is precisely this openness to variation that allows us to observe how Liraglutide’s effects may differ across contexts. In qualitative research, depth of understanding is prioritized over statistical control; transferability of findings then depends on providing rich descriptions so that readers can judge applicability to other settings. To that end, we have added more contextual detail in the revised manuscript (e.g., participants’ baseline psychiatric profiles, medication regimens, and types of therapy received) so that others can assess how our themes might resonate with their own patients’ situations.

-Foundation for future research:

We fully agree that larger samples, pre‐ and post‐treatment assessments, and long‐term follow‐up are essential next steps. Our intention is that the patterns and patient‐reported experiences identified here will inform the selection of standardized outcome measures (e.g., validated scales for emotional dysregulation, binge frequency, and quality of life) and guide selection criteria for participants in subsequent trials. We have clarified in the Discussion how our qualitative findings can be translated into specific, testable hypotheses for quantitative studies.

We hope that the revisions address your concerns and underscore the value of qualitative evidence in shaping future BED research.

Reviewer 3:

1. Comment: The abstract gives a clear summary of the study, covering the background, methods, results, and key findings. It explains that the study looks at how people with obesity and BED experience using Liraglutide. However, it doesn’t mention the number of participants (8), and it only says the study is qualitative halfway through the abstract. This could confuse readers who might think it’s a clinical trial. It would be better to state early on that it’s a qualitative study using thematic analysis and include the sample size upfront.

• Response: Thank you for this observation. We agree that stating the study design and, in the abstract, would provide greater clarity and help prevent misinterpretation. In response, we have revised the abstract to explicitly state that this is a qualitative study. These changes are intended to ensure that readers immediately understand the nature and scope of the study. Please see the revised abstract in the updated manuscript.

2. Comment: The introduction gives useful background information and explains how BED and obesity are related. It also explains why Liraglutide might help. However, the section is a bit too long and repeats similar ideas about emotional eating, dieting, and weight concerns. Also, the main research aim is only mentioned at the end. The introduction would be clearer if the authors explained the study's goal earlier and removed repeated points to make it easier to read.

• Response: We are thankful for your helpful feedback. We appreciate your suggestion to move the aim of the study earlier in the introduction and to reduce repetitive content for better clarity. We agree that clarity is important. However, it is common to present the research aim toward the end of the introduction—after establishing clinical background, reviewing relevant literature, and demonstrating the research gap. This structure allows the aim to emerge naturally from the problematization and ensures the reader fully understands the context and rationale behind the study before encountering the research objective. That said, we have revised the introduction to tighten the language, reduce repetition, and ensure that the research aim is clearly presented and logically placed at the end of the introduction. We believe this maintains the flow while making the aim and rationale more prominent and easier to follow.

3. Comment: This section clearly describes how the study was done, including how participants were chosen, what kind of interviews were used, and how the data were analyzed. It also covers the ethical steps taken. However, it doesn’t say whether different researchers checked each other’s coding or how they made sure the themes were reliable. It would be helpful to include information on how they checked for consistency—like if more than one researcher coded the data or if they discussed themes together to make the findings more t

---

## [Decision Letter · Decision Letter 1]

17 Oct 2025

Patient experiences with liraglutide for obesity and binge eating disorder

PONE-D-24-56405R1

Dear Dr. Folling

We’re pleased to inform you that your manuscript has been judged scientifically suitable for publication and will be formally accepted for publication once it meets all outstanding technical requirements.

Kind regards,

Saeed Ahmed, MD, FAPA, FASAM

Academic Editor

PLOS ONE

Additional Editor Comments (optional):

I am pleased to inform you that your manuscript entitled “Patient Experiences with Liraglutide for Obesity and Binge Eating Disorder” has been accepted for publication following peer review.

The reviewers and editorial board found your study to be an important contribution to the growing body of research on pharmacological interventions for obesity and associated eating disorders.

The manuscript will now proceed to our production department for copyediting and typesetting. You will receive proofs for your review in due course. Please ensure that all author details and affiliations are correct at that stage.

Reviewers' comments:

Reviewer's Responses to Questions

**Comments to the Author**

1. If the authors have adequately addressed your comments raised in a previous round of review and you feel that this manuscript is now acceptable for publication, you may indicate that here to bypass the “Comments to the Author” section, enter your conflict of interest statement in the “Confidential to Editor” section, and submit your "Accept" recommendation.

Reviewer #2: All comments have been addressed

Reviewer #3: All comments have been addressed

2. Is the manuscript technically sound, and do the data support the conclusions?

Reviewer #2: Yes

Reviewer #3: Yes

3. Has the statistical analysis been performed appropriately and rigorously? 

Reviewer #2: N/A

Reviewer #3: N/A

4. Have the authors made all data underlying the findings in their manuscript fully available?

Reviewer #2: Yes

Reviewer #3: No

5. Is the manuscript presented in an intelligible fashion and written in standard English?

Reviewer #2: Yes

Reviewer #3: Yes

6. Review Comments to the Author

Reviewer #2: This article provides a thorough overview of the relationship between obesity and Binge Eating Disorder (BED), discussing both psychological and physiological factors and exploring treatment options such as psychotherapy, pharmacotherapy, and the use of Liraglutide. The section presents a well-developed thematic synthesis that links emotional regulation in BED to participants’ experiences with Liraglutide. However, the study has several shortcomings, including a small sample size, limited reliability, selection bias, and an inability to compare pre- and post-treatment outcomes or use longitudinal models.

Reviewer #3: The authors have addressed all my comments thoughtfully and thoroughly. The abstract now clearly states the qualitative design and sample size upfront. The introduction has been tightened to reduce repetition and improve flow, while keeping the aim in a logical position. The methods section now explains how coding and theme development were checked for consistency. The results remain well supported with participant quotes and a clear thematic figure. The discussion has been revised to reflect the small-sample study and avoid overgeneralization.

7. PLOS authors have the option to publish the peer review history of their article (what does this mean? ). If published, this will include your full peer review and any attached files.

**Do you want your identity to be public for this peer review?** For information about this choice, including consent withdrawal, please see our Privacy Policy .

Reviewer #2: No

Reviewer #3: No

---

## [Editor Report · Acceptance letter]

PONE-D-24-56405R1

PLOS ONE

Dear Dr. Følling,

I'm pleased to inform you that your manuscript has been deemed suitable for publication in PLOS ONE. Congratulations! Your manuscript is now being handed over to our production team.

Kind regards,

on behalf of

Dr. Saeed Ahmed

Academic Editor

PLOS ONE